# Critical Factors Influencing Early Contract Termination in Public Design–Build Projects in Developing and Emerging Economies

Claudia Riveros [1], Angie L. Ruiz [2], Harrison A. Mesa [1,3,*] and Jose A. Guevara [2]

1 School of Civil Construction, Faculty of Engineering, Pontificia Universidad Católica de Chile, Santiago 7820436, Chile; cmriveros@uc.cl
2 Department of Civil and Environmental Engineering, Universidad de los Andes, Bogota 111711, Colombia; al.ruiz10@uniandes.edu.co (A.L.R.); ja.guevara915@uniandes.edu.co (J.A.G.)
3 National Excellence Center for the Timber Industry (CENAMAD), Pontificia Universidad Católica de Chile, Santiago 7820436, Chile
* Correspondence: hmesa@uc.cl

**Abstract:** Public projects continually face multiple difficulties for their satisfactory completion. One of the most complex challenges is early contract termination (ECT), which delays social goods delivery and exhausts public resources. This study aimed to determine the root causes of the critical factors that lead to ECT in public building projects. We studied 20 kindergarten construction projects in Chile through a multi-case study. It addressed a pattern-matching analysis of symptoms associated with risks of design–build (DB) contracts and a five whys analysis to determine the root causes of the symptoms identified in the units of analysis. The results show that ECT projects' most common symptoms are labor force shortage, materials shortage, and non-payment claims. In addition, the root-cause analysis exposed that the main causes of ECT's symptoms were deficiencies in the bidding evaluation process, which led to an inadequate selection of the general contractor, lack of experience of the owner, and regulatory limitations of the legal framework for public projects. The construction projects faced ECT and cost and time overruns associated with poor risk management due to the owner's and general contractor's lack of experience in DB contracts.

**Keywords:** early contract termination; DB contracts; risk management

## 1. Introduction

One of the public administrators' biggest challenges is how to complete public projects within the time, cost, and quality standards [1]. Development of public projects is a high-risk activity due to the complex processes, long periods, uncertain environment, and financial requirements associated with construction contracts [2]. The risks present in public projects are significantly influenced by the contracting strategy selected [3]. It has become common within contracting strategies to integrate the design and construction stages into one contract. This method is known as the design–build (DB) system and has become popular, especially in developed countries [4,5]. On the other hand, this strategy has not been formalized in developing countries. However, the type of contract used by public agencies in public projects has characteristics attributable to DB contracts.

Through the DB method, public agencies assign both design and construction to a single company [6]. The general contractor assumes most of the project's risks since it is responsible for the design and construction. According to Oluwaseyi Modupe et al. [1], general contractors in developing countries usually do not have the experience required to manage risks effectively. The lack of DB experience in both the public and the private sectors leads to a myriad of problems (e.g., cost overruns, revenue deficiency, and construction schedule delays), which can result in an early termination of the contract.

Early contract termination (ECT) implies the suspension of projects and is one of the public sector's most complex problems [7]. In the context of this research, early termination means to end the contract between the owner and the general contractor without concluding the design or construction activities of the project under development before the contract deadline. This situation causes a temporary suspension of the design or construction activities; meanwhile, the owner (public agency) finds a new general contractor. Typically, early termination of a contract occurs because of the parties' impossibility to comply with the contract (for example, due to a prolonged event of force majeure) or default events such as insolvency of one of the parties [8]. As previously mentioned, ECT, especially in public projects, significantly affects the economic resources and social well-being of a country. In addition, ECT generates losses of resources and damages for the public administrator considering the deterioration of public investment and the delay in delivering a property destined for a social benefit [9,10]. Numerous construction projects have been "abandoned" worldwide, and many researchers have analyzed the consequences of this for the parties involved [9,10]. Although early termination of construction contracts is a usual issue in public projects in developing countries, few extant studies of this phenomenon identify and analyze the factors that lead to ECT.

Some scholars have stated that ECT evidences the lack of proper risk management in developing countries [11]. Therefore, to analyze ECT, it is necessary to investigate the risks associated with construction contracts. Based on the literature review's evidence about the lack of experience of developing countries in implementing DB contracts and managing risks effectively, this research focused on studying ECT in this type of contract.

Some researchers have determined and classified the main risks associated with DB contracts. Specifically, based on previous research, the following risk types are identified: scope risks, third-party and complexity risks, construction risks, utility risks, design and contract risks, and management risks [12,13]. Similarly, other authors have identified the potential sources of risks, including lack of planning, incorrect cost estimates, poor design, political factors, and inadequately defined roles and responsibilities [2,14–16]. However, most studies that analyze risks focus on examining those associated with time and cost overruns, leaving aside other important consequences like ECT.

This study aimed to determine the critical factors and their root causes in public building projects that lead to ECT in DB contracts. To fulfill this objective, a multi-case study was proposed to analyze a set of public building projects (construction of nurseries and kindergartens) and answer the following question: why does a public building project result in ECT? Pattern matching and root-cause analysis techniques were employed to analyze the common symptoms among the units of analysis, and the possible causes associated with them were analyzed. The results show that the root causes of the critical factors that affect the early termination of contracts are closely related to selecting the ideal general contractor for the contract type, poor risk management, and lack of experience of the general contractor and the owner. Based on that, recommendations are provided to avoid ECT in future projects.

This research adds to the body of knowledge in construction management and risk management research and will help the AEC industry in developing countries understand the challenges and causes of ECT in public building projects developed with DB contracts. Furthermore, identifying the root causes of the critical factors (symptoms) that lead to the early termination of this type of contract allows public administrators to directly address the root causes and not superficial problems to improve the design and management of DB contracts in developing countries.

## 2. Background

ECT remains a recurring problem in developing countries [10,17,18]. Developing countries have a reputation for developing failed construction projects, with time and cost overruns and ECT in worst-case scenarios. Some researchers consider that this is because of the lack of efficiency in risk management in developing countries [1]. In other words, ECT

is a manifestation of the risks present in the development of projects. For this reason, as a starting point in this study, relevant research related to the risks of construction contracts and their consequences was carefully reviewed, specifically, the set of risks associated with DB contracts, which fit the case study developed in this research.

It is not easy to generalize all the potential risks associated with construction contracts since they depend significantly on a project's context. However, it is possible to identify common categories of risks within different research works. Typically, risks are classified into the following groups: scope risks, third-party and project complexity risks, design and contract risks, right-of-way and utility risks, construction risks, and management risks [12,13]. Scope risks are all risks associated with project definition, scope definition, staff experience, and conformance with regulations and documentation [13]. Third-party and complexity risks include concerns related to project complexity, approval by different agencies, legal challenges, and delays associated with utility agreements [8]. Design and contract risks are related to the percentage of design completion before the contract is awarded. Depending on the initial design specifications, design changes may be needed, which will affect the work progress [14]. Right-of-way and utility risks include risks related to access or work areas where construction activities are carried out [13]. Construction risks refer to the uncertainties of construction activities, geotechnical investigation, and environmental impact [14]. Finally, management risks include project and program management and insurance; poor project management can lead to project delays [12].

Scholars have examined the main factors influencing the different types of risks, such as risk categories. The factors are usually grouped into political, environmental, financial, managerial, and external factors [2,11,14]. For example, Chan et al. [19] highlighted as critical factors the lack of competencies of contractors and clients, the lack of the project team promise, restrictions on the project development imposed by final users, and the lack of understanding of risk and liability assessments. Other authors have recognized other factors such as inadequate financing, inflation, inadequate cost control, improper documentation, unqualified/inexperienced consultants, persistent community eruptions and interference and disputes, and natural disasters [11,17].

Successful development of projects depends on proper risk management. For this, it is necessary to identify risk factors; the more risks are identified in the initial stages of projects, the greater the chances of achieving the projects' objectives [1]. Therefore, identifying risks is vital to avoid ECT, especially in developing countries where risks are increased because of poor risk management, poor quality control by regulatory agencies, corruption, and inconsistent government policies [11].

This study aimed to determine the risks and factors that lead to ECT based on the above. All types of risks are associated with potential consequences or issues like design modifications, material changes, construction rework, decreased profits, insufficient labor force, delay in the general contractor's financial payments, and time and cost overruns. This set of consequences is usually easy to track in the development of projects as evidence of the manifestation of risks. Therefore, the diagnosis conducted in this study is based on identifying these potential consequences in practice. Table 1 summarizes the identified main consequences, representing the theoretical symptoms for this study. As explained before, the ECT condition is a consequence of poor risk management; therefore, as a starting point to find ECT causes in public building projects, the authors selected the most common risks of construction contracts and the consequences associated with them.

**Table 1.** Theoretical symptoms of risks.

| Type of Risks | Theoretical Symptoms | 1 | 2 | 3 | 4 | 5 | 6 | 7 | 8 | 9 | 10 | 11 | 12 | 13 | 14 | 15 | 16 | 17 | 18 | 19 | 20 |
|---|---|---|---|---|---|---|---|---|---|---|---|---|---|---|---|---|---|---|---|---|---|
| Construction risks | Labor force shortage | X | | X | X | X | | X | X | X | X | | X | X | X | | | X | X | X | X |
| Construction risks | Non-payment claims | X | X | | | X | | | | | X | X | X | | | X | X | | | | X |
| Construction risks | Materials shortage | X | | X | X | X | X | X | X | X | X | X | X | X | X | | X | X | X | | X |
| Financial risks | Lack of advance payments | | X | | | | X | | | | | | X | | | | | | | | |
| Financial risks | Delayed payment by the client | X | | X | X | X | | X | | | X | X | X | X | X | X | X | X | | | X |
| Management risks | Erroneous information about electricity and water service | | | | X | | | X | | | X | | | X | | X | X | | | X | |
| Management risks | Slowness of the owner's decision-making process | X | | | | X | | X | | | X | | X | X | | | | | | | X |
| Management risks | Irregularities in the land's background | X | | X | X | X | X | X | X | | X | X | X | X | X | | X | X | | | X |
| Management risks | Poor supervision | X | | | | X | | X | | | X | | X | X | X | | | X | | | |
| Management risks | Lack of communication between the parties involved | | | | | X | | X | | | X | | | X | X | | | | | | X |
| Management risks | Community opposition | | | | X | | | | | | | | X | | X | | X | X | | X | |
| Design and contract risks | Design changes | X | X | X | X | X | X | | X | X | X | | X | X | X | X | | X | X | X | |
| Design and contract risks | Conflicts between the general contractor and the owner | X | | | | X | | | | | | | | X | | | | | | | |
| Third-party and complexity risks | Delay in getting a building permit | X | | X | | X | | X | | | X | X | X | X | | X | | X | | X | |
| Other risks | Weather effects | X | | | | X | | X | | | X | X | X | X | X | | | X | | | |
| Other risks | Construction accidents | X | | | | | | | | | X | | X | X | | | | X | | | |

Note: 1 = [20]; 2 = [2]; 3 = [21]; 4 = [22]; 5 = [23]; 6 = [9]; 7 = [24]; 8 = [25]; 9 = [26]; 10 = [27]; 11 = [18]; 12 = [12]; 13 = [28]; 14 = [14]; 15 = [29]; 16 = [11]; 17 = [1]; 18 = [30]; 19 = [13]; and 20 = [31].

## 3. Methodology

We analyzed the critical factors influencing ECT in public building projects through the case study method. Figure 1 shows the methodology followed to accomplish the objectives of this study. The first step was to select case studies and their units of analysis, and the second one was the data collection for each unit of analysis. Then, the pattern-matching technique was used to identify matches and mismatches between the cases evaluated. The next step was to develop a root-cause analysis to establish the main causes of the early termination of contracts. Finally, based on the results, recommendations to improve the regulation of contracts are given to avoid early contract termination.

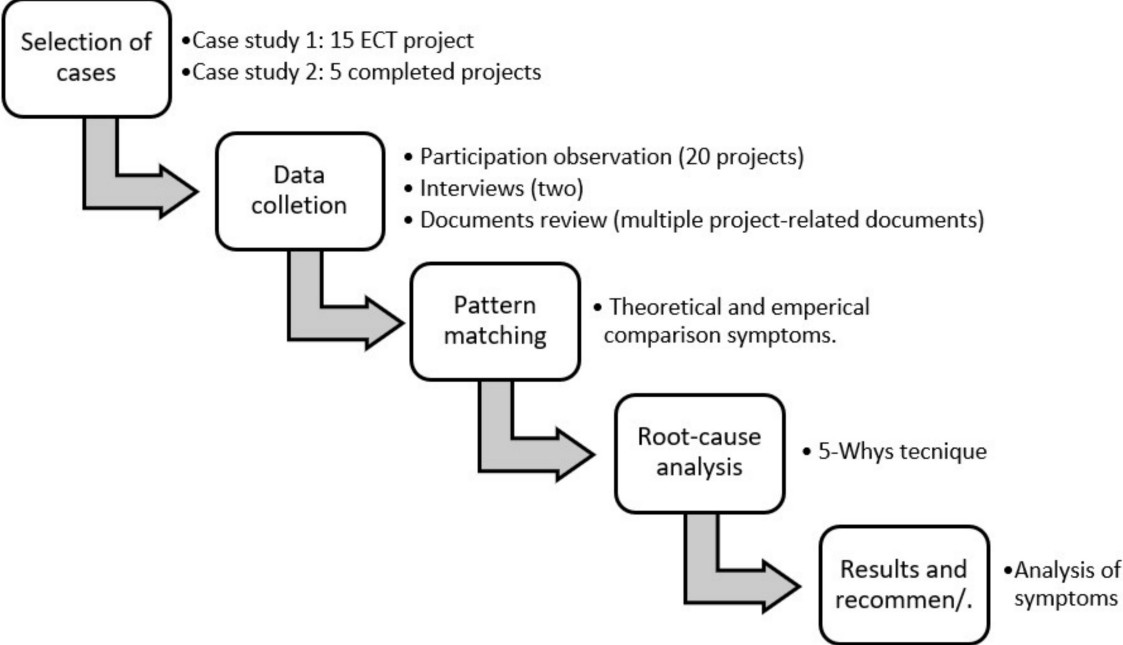

**Figure 1.** Methodology.

This investigation focused on building construction projects in Chile. Chile was selected for the case study for three main reasons: (1) it is a developing country, and ECT frequently occurs in these countries, as mentioned before; (2) despite being a developing country, Chile presents one of Latin America's fastest-growing economies, which allows the country to develop high-quality projects, and (3) data and documentation about projects is publicly available or can be easily accessed on request, which contributes to the consistency of the data employed. This study analyzed 20 projects of the program "Aumento de Cobertura" of the National Kindergartens Board (JUNJI), an organism of the Ministry of Education. We analyzed the type of contract and the instruments JUNJI used to develop nurseries and kindergartens. The program used a design–build contract with lump sum and unit price as compensation methods. According to the procurement documents, the general contractor was in charge of the engineering, construction, and assembly of all the project facilities and the direction, integration, and coordination of design and construction [6].

Standard tendering documents were defined for all projects, and this group of documents regulates bidding and contracting processes [32]. The contract's fulfillment is supervised by the Technical Inspection of Works (ITO), which supervises that fulfillment of the contract and the technical specifications is timely and in good faith. The ITO also gives instructions related to the work's execution, oversees workers' safety conditions, labor, and social security obligations, and indicates to the general contractor the problems that may affect the contract's development.

For the projects analyzed in this study, ECT is defined as the end of a design–build contract without concluding the work in the design or construction stage before the stipu-

lated contract deadline because one or both parties present one of the causes established in Public Procurement Law 19.886, art. No. 77, during the project development [33]. This law establishes that public contracts can be modified or terminated early for the following reasons: (1) mutual agreement between the contracting parties; (2) a serious breach of the obligations contracted by the general contractor; (3) insolvency of a contracting party (unless the guarantees for fulfillment of the contract are provided); (4) public interest or national security reasons; (5) specific causes that are included on the tender documents or the contract; and (6) dissolution of the general contractor company (without a legal successor) or death of the general contractor in the case of a natural person. The following are considered serious non-compliance by the general contractor: delay of more than five days at the beginning of the design or the execution of the project, repeated non-compliance with the observations of the instructions given by JUNJI or ITO, non-compliance with the obligation of confidentiality, and repeated delay in the delivery of reports or development of the work set for each stage.

### 3.1. Selection of Cases

This study presents a multiple-case study approach, where two cases are evaluated, both with multiple units of analysis. The first case study focused on analyzing the characteristics of projects that presented early termination of the contract (ECT projects). That is, these projects were under development in the construction stage, and the DB contract between JUNJI and the general contractor was ended before the contract deadline without concluding the construction activities. On the other hand, the second case study examined projects with time and cost overruns where design and construction were completed and the Ministry of Education accepted them. Therefore, the second case study was selected to analyze why these projects were completed and did not result in ECT even though they presented time and cost overruns. Hence, this analysis allowed identifying the symptoms in the completed projects and compare them with the ECT projects' symptoms. If they were the same, we would identify why the contract was not terminated early in these cases.

The program comprises 469 projects, of which 327 have been finalized, 100 presented early termination of the contract, and the remaining are in execution. For the first case study, 15 projects (one for each region) were randomly selected among the 100 projects that presented an early contract termination, i.e., the projects that have the resolution of early termination by the Comptroller General of the Republic. Additionally, five projects were randomly selected among the 327 completed projects as the units of analysis of the second case study. These projects were chosen in the regions where there was more than one case of contracts with early termination. Table 2 shows the list of the projects selected for the analysis.

**Table 2.** Units of analysis.

| Case Study | Units of Analysis | National Region | Project Name |
|---|---|---|---|
| | 1 | Región de Tarapacá | La Pampa Alto Hospicio |
| | 2 | Región de Antofagasta | Los Chungungos |
| | 3 | Región de Atacama | Nva Castilla |
| | 4 | Región de Coquimbo | Vial Recabarren |
| | 5 | Región de Valparaíso | Tierras Rojas |
| | 6 | Región de Libertador Bernardo O'Higgins | La Gamboina |
| | 7 | Región de Maule | Alquihue |
| Case study 1: ECT projects | 8 | Región de Bío-Bío | Miguel Astroza |
| | 9 | Región de Araucanía | Puerta Sur |
| | 10 | Región de Los Lagos | Comites de Vivienda |
| | 11 | Región de Aysen | Costanera Punta Arenas |
| | 12 | Región de Magallanes y la antártica chilena | Augusto Dhalmar |
| | 13 | RM Región Metropolitana de Santiago | Sector Rural de Liquiñe |
| | 14 | Región de los Rios | Lomas de Machalí |
| | 15 | Región de Arica y Parinacota | Costa del sol |

**Table 2.** *Cont.*

| Case Study | Units of Analysis | National Region | Project Name |
|---|---|---|---|
| | 16 | Región de Tarapacá | Huantajaya |
| | 17 | Región de Coquimbo | Los Ruiles |
| Case study 2: Completed projects | 18 | Región de Libertador Bernardo O'Higgins | Adriana Lyon |
| | 19 | Región de Maule | Don Rodrigo |
| | 20 | Región de Magallanes y la antártica chilena | Cumbres Patagónicas |

*3.2. Data Collection*

It is essential to collect data from different sources in qualitative research to ensure reliability and validity. For this reason, this study, in addition to the literature and documentation review, is supported by information obtained through participant observation and interviews with key informants.

Participant Observation and Interviews

Observations are a valuable source of information for case studies since they take place in the real world. In case studies, where the problem is not purely historical, some relevant context and environmental conditions are available for observation [34]. The first author of this study is a JUNJI member and was present in some of the projects considered in the analysis, specifically in the projects represented by the units of analysis 1, 6, 7, 8, 9, 14, and 15 (all of them presented ECT). When the researcher is not merely a passive observer, the observation mode is known as participant observation. This means that the researchers assume roles requiring their participation in the studied actions [35].

The first author has been a member of JUNJI for four years. Her functions include reviewing project design (in terms of architecture and regulations), contract monitoring and authorization of their modifications in specific country regions, and implementing strategies with the teams when critical issues arise. Additionally, the first author coordinated meetings to review the budget execution progress committed to the Ministry of Finance and visited the projects to review their progress rate. The first author also had the opportunity to witness the interactions and decision-making processes involved in the project execution. The above gives the author the experience and knowledge to analyze the objectives of this study. Furthermore, her direct relationship with the projects and the contact with other actors involved in them allow her to know the context and reasons behind the projects' difficulties, thus facilitating the analysis of the causes of the possible consequences identified in this research.

Due to the ease of meetings between the first author and other project stakeholders, interviews were carried out to strengthen the analysis. Interviews are usually adopted as a source of information in case studies. This technique seeks to access the project participants' information by scanning their profiles and questions prepared in advance. Following this, the interviews are transcribed for further analysis. The first author of this study is a participant in the projects; however, interviews with other participants were carried out to integrate different perspectives.

Specifically, two key informants were interviewed: the legal coordinator and the project coordinator. They were selected because they were in charge of coordinating the writing of the contract documents and meeting with other government entities (e.g., Ministry of Public Works and Ministry of Housing) with more experience in developing construction projects to collect information and thoughts to design the contract. They also met with the National Comptroller's Office to coordinate the revisions and approvals to the final contract document. Two interviews were conducted for each key informant, and they took an average of 60 min.

The interviews' main objective was to identify the reasons for choosing a design–build contract (lump sum and unit price) for the works of the program "Aumento de Cobertura." The interview sought to learn from those who designed the tender documents why this type

of hiring was considered and the problems identified from the practical point of view to compare the information with the literature review results and the documentation analysis of the case studies. That is why the interviews were focused on knowing how the tender documents were formulated and if it was considered how the problems described in the literature for this contract type would be addressed, specifically risks associated with the distribution of responsibilities.

### 3.3. Document Review

The first author had access to a suite of project-related documents. Different instruments that define the contractual reasons for an early contract termination were analyzed to establish qualitative parameters and identify critical factors. Usually, the factors that lead to ECT consequences are preceded by symptoms described by construction professionals in weekly progress reports and observations of works through a management and monitoring system. Besides, financial reports are considered for this analysis to know if problems regarding the payment system influenced the early termination (e.g., delay of payment by the owner or lack of advance payment that may have brought consequences). The formal collection and organization of these documents for all projects took approximately five months. However, it is worth clarifying that the researcher has observed and analyzed the problem before starting this work.

#### 3.3.1. Case Study 1: Early Contract Termination

A review of the documentation that technically and contractually determines the early contract termination was carried out. The following documents were reviewed:

- Weekly report of the Technical Inspection of Works: this document records information regarding the last physical percentage of work progress registered, characteristics of the progress curves (scheduled vs. actual), and an evaluation of the reports six weeks before early termination.
- Administrative resolution of early contract termination: this document describes the ITO's reasons for justifying the early termination of the contract.
- Technical reports of ITO (other antecedents): complementary ITO's reports that provide alerts on anomalies in the work's progress. These documents contain technical observations detected by ITO that seek to alert serious breaches of the contract.
- Construction finance reports: these documents allow the authors to identify the frequency and amount of the payment statements and contract modifications. The main purpose of reviewing these reports is to identify if the project had an advance payment and adequate financial liquidity.

#### 3.3.2. Case Study 2: Completed Projects

During the development of the completed projects, different symptoms were identified to analyze the relationship between these projects and those that presented ECT. This analysis allowed the authors to determine if the completed projects showed the same symptoms as those with early contract termination. If the symptoms between the ECT projects and the completed projects matched, the analysis was focused on why these projects were completed. The following documents were reviewed:

- Technical reports were reviewed to verify the existence of penalties and the reasons that support them, which are usually associated with breaches of the contract.
- The contractor's discharges provide information to identify the claims' origin and their relationship to the ECT projects' symptoms.
- Contract modifications provide information about modifications regarding time and costs increase or decrease or extraordinary work. These documents were reviewed to analyze if the projects were free of problems during their execution.

### 3.4. Pattern Matching

Pattern matching logic is one of the most common techniques employed for the qualitative analysis of case studies. It compares a theoretical or predicted pattern (based on the literature review) with an empirical pattern based on the case study's findings [34]. The main purpose is to decide if the observed pattern matches the expected one. For this study, the symptoms identified in both case studies (projects with ECT and completed projects) were compared with the theoretical symptoms summarized in Table 1. Additionally, the results of the two case studies were compared to strengthen the analysis through rival explanations. This is used when it is known there was a particular outcome in each case study (ECT or completed) and the research is focused on understanding why these outcomes occurred in each case.

### 3.5. Root-Cause Analysis

The five whys technique was employed to identify the causal roots of the symptoms identified through the literature review and pattern matching. This technique is an iterative process that involves repeating multiple times to help identify the root cause of the problem and its solution [36–38]. This method has been employed in different fields; for example, Tsao et al. [39] applied five whys to develop local and global fixes for a system of precast walls and door frames. Yeganeh et al. [40] used this technique to identify design–construction interface problems and their causes in design–build projects. We applied five whys to find the causes related to each undetermined symptom, establish whether the same cause is responsible for several symptoms, and define whether they correspond to critical factors that lead to ECT. For this, we intended to find the reason for the occurrence of each symptom, why this first cause was identified, and so on three more times until reaching the root cause.

### 3.6. Validity and Reliability

In this research, different validity and reliability strategies were considered. First, multiple data sources were employed to construct validity, and draft results were reviewed with the key participants [41]. Second, a multiple case study, including contrasting case studies, was employed to compare and increase data accuracy to strengthen the analysis. The second author reviewed this process to identify the matches between the potential theoretical symptoms and those in practice to control the pattern matching reliability. This process review strengthened the pattern matching results, supported by an agreement of 85%. Additionally, the third and fourth authors iteratively reviewed the pattern matching and root-cause analysis results for both case studies until reaching a consensus.

## 4. Results

This study used pattern matching to compare the theories about design–build contracts' symptoms to what has happened in practice in construction projects. Table 3 shows the theoretical symptoms identified in the analysis units. Those symptoms that were identified in the literature but not in the case studies are not presented in the results. For case study 1, the following are the symptoms identified and the percentage of projects in which they were presented: (A) labor force shortage (86.6%); (B) non-payment claims (73.3%); (C) materials shortage (53.3%); (D) lack of advance payments (20%); and (E) delayed payment by the client (13%). Specifically, to get ECT, Table 3 shows that 13% of the projects presented only one symptom (A)–(C) (units of analysis 7 and 11), 40% presented two symptoms (A)–(C) (units of analysis 3, 5, 10, 12, 13, and 14), 40% of the sample had three symptoms (A)–(E) (units of analysis 1, 2, 4, 6, 8, and 9), and the unit of analysis 15 presented all the five symptoms.

**Table 3.** Pattern matching results.

| Symptoms | Case Study 1 | | | | | | | | | | | | | | | Case Study 2 | | | | |
|---|---|---|---|---|---|---|---|---|---|---|---|---|---|---|---|---|---|---|---|---|
| | **1** | **2** | **3** | **4** | **5** | **6** | **7** | **8** | **9** | **10** | **11** | **12** | **13** | **14** | **15** | **16** | **17** | **18** | **19** | **20** |
| (A) Labor force shortage | X | X | X | X | X | X | | X | X | X | X | X | | X | X | | | | | |
| (B) Non-payment claims | X | X | X | | X | X | X | X | X | | | | X | X | X | | | | | |
| (C) Materials shortage | X | | | X | | | | X | X | X | | X | X | | X | | | | | |
| (D) Lack of advance payments | | X | | X | | | | | | | | | | | X | | | | | |
| (E) Delayed payment by the client | | | | | | X | | | | | | | | | X | | | | | |
| (F) Erroneous information about electricity and water service | | | | | | | | | | | | | | | | X | | X | | |
| (G) Irregularities in the background of the land | | | | | | | | | | | | | | | | X | | | X | |
| (H) Community opposition | | | | | | | | | | | | | | | | | | | X | |
| (I) Design changes — (I.1) due to change in regulations | | | | | | | | | | | | | | | | | | | | X |
| (I) Design changes — (I.2) by owner's request | | | | | | | | | | | | | | | | X | X | | | |
| (I) Design changes — (I.3) due to mechanical and topographical information | | | | | | | | | | | | | | | | X | | X | | X |
| (I) Design changes — (I.4) due to problems with soil quality | | | | | | | | | | | | | | | | | | X | | X |
| (J) Delay in getting a building permit | | | | | | | | | | | | | | | | X | X | | X | X |

Table 3 also shows that the set of symptoms (F)–(J) were exclusively identified in case study 2. The symptom (I) is presented in four subcategories since the revised documents specified the reason behind the design changes. Specifically, 20% of the units of analysis of case study 2 presented (E) delayed payments by the client; 40% of the sample presented (F) erroneous information about electricity and water service, (G) irregularities in the background of the land, (I.2) design changes on owner's request, and (I.4) design changes due to problems with land quality; 60% of the sample presented (I.3) design changes due to mechanical and topographical information; and 80% of the projects presented (J) a delay in getting a building permit.

The documentary review showed that all the completed projects presented time and cost overruns. Specifically, the units of analysis 16, 19, and 20 took 143%, 148%, and 133% of the contract's time, respectively. Units of analysis 18 and 19 required 230% and 210% of the initial time for completion, respectively. Additionally, all the projects presented variations of the original contract value; these variations ranged between 3% and 28%. The same unit of analysis 19 presented the greatest variation in time and costs.

*4.1. Interviews*

The interviewees indicated JUNJI used a design–build contract with lump sum and unit price as compensation methods to manage the quantities' and costs' uncertainty of the construction activities (e.g., excavations and foundations) because the projects were tendered using a schematic design without soil's topographic and mechanical information. The same contract was designed for all the nurseries and kindergarten projects in the country to speed up the works' execution, modifying only the project's particularities. The interviewees stated that the priority was hiring and executing the projects as quickly as possible. For this reason, no strategies were developed to address the risks and their consequences that could arise in the project execution from a technical perspective.

Additionally, the interviewees indicated that the National Comptroller's Office requested that the bidding process should be available to the greatest number of bidders since JUNJI does not have a contractor's registry compared to other government institutions (e.g., the Ministry of Public Works). As for weaknesses, the interviewees identified that the general contractor selection process was not rigorous. The contractual documents had an evaluation methodology that did not allow JUNJI to adequately corroborate the general contractor's economic capacity and experience. Furthermore, JUNJI did not have an adequate system to evaluate the general contractor's performance after finishing the project. Hence, JUNJI could not reject their participation in other bidding processes if the general contractor presented deficiencies in their performance. Besides, they indicated that there was no adequate mechanism to control the distribution of the project budget, incentives, and payments of the general contractor.

*4.2. Root-Cause Analysis*

The information from the observations registered in ITO's weekly reports and technical reports was considered to identify the symptoms that occurred in each project and analyze them to identify the root causes. In addition, the financial reports were reviewed to determine anomalies in the contract payments, or if there was no advance, and if these could impact the problems that arose in these projects. Based on these, relationships were established between the identified causes and the main symptoms in the projects that were identified in the previous section.

Table 4 synthesizes the causal relationships identified through a five whys analysis for the main symptoms identified in the literature review and case study 1 evaluation. Once the symptom was identified, the authors asked and answered why it occurred at least five times in succession until identifying an actionable root cause. For example, why did the labor force shortage occur? The labor force shortage occurred because the project lacked financial resources. Why did the project lack financial resources? The project lacked financial resources because of the general contractor's financial insolvency. Why

did the general contractor present financial insolvency? The general contractor presented financial insolvency because of poor financial resource management. Why did the general contractor have a deficiency in financial resource management? The deficiency of financial resource management occurred because of the general contractor's lack of experience in this type of contract. Why did the general contractor have lack of experience? The general contractor's lack of experience in this type of contract occurred because of the deficiencies in the bidding evaluation process to select an adequate general contractor to manage a design–build contract.

**Table 4.** Root-cause analysis results of case study 1.

| Symptoms | Why 1 | Why 2 | Why 3 | Why 4 | Why 5 |
|---|---|---|---|---|---|
| (A) Labor force shortage | Lack of resources. | General contractor's financial insolvency. | Financial resources management deficiency. | Lack of experience (general contractor). | Deficiencies in bidding evaluation. |
| (B) Non-payment claims | Lack of resources. | General contractor's financial insolvency. | Financial resources management deficiency. | Lack of experience (general contractor). | Deficiencies in bidding evaluation. |
| (C) Materials shortage | Lack of resources. | General contractor's financial insolvency. | Financial resources management deficiency. | Lack of experience (general contractor). | Deficiencies in bidding evaluation. |
| (D) Lack of advance payments (financing) | Delay in the disposition of funds. | Delay in requesting funds | Lack of clarity in the amounts requested from the Ministry of Social Development (MSD) | Lack of neatness in the request for funds to the MSD. | Lack of experience (JUNJI). |
| (E) Delayed progress payment by the owner | Poor financial evaluation of the general contractor. | Lack of experience (general contractor). | Deficiencies in bidding evaluation. | Limitations in contract provisions. | Regulatory limitations of the legal framework for public projects. |

From the analysis of the five whys, the following are identified as causes of these potential symptoms: (1) deficiencies in the bid evaluation process, which reflects the lack of experience of the general contractor; (2) lack of experience of JUNJI in building projects; and (3) the regulatory limitations of the legal framework for public projects. Cause (1) is related to the general contractor and is supported by symptoms (A) labor force shortage, (B) claims of non-payments, and (C) materials shortage. On the other hand, causes 2 and 3 are related to JUNJI's governance, supported by the symptoms (D) lack of advance payment and (E) delayed progress payment by the owner.

Table 5 shows the results for case study 2 of the root-cause analysis. The following causes were identified: (1) JUNJI's lack of experience; (2) regulatory limitations of the legal framework for public projects; and (3) limitations of external entities. From these root causes, the problems directly related to JUNJI's lack of experience were the (I) design modifications due to (I.1) changes in government regulations and (I.2) on request of the owner. The cause of "regulatory limitations of the legal framework for public projects" is related to the problems that were produced by design changes given the (I.3) information on soil mechanics and topographies, which was presented in 60% of the sample, and the modifications (I.4) due to the quality of the soil, identified in 40% of the projects. Finally, the cause of "limitations of external entities" is related to (F) erroneous information about electricity and water service; (G) irregularities in the background of the land, (H) community opposition; and (J) delay in getting a building permit.

Additionally, it was identified how the general contractor and the owner assumed the risk associated with each symptom. This was completed based on the description of the risk and the contract's clause or section within the procurement documents to which it was associated. In case study 1, the results show that the general contractor assumed 100% of the risks in the construction stage (Table 6). For case study 2, it was found that 75% of the problems associated with these risks were the owner's responsibility (Table 7).

**Table 5.** Root-cause analysis results of case study 2.

| Symptoms | Why 1 | Why 2 | Why 3 | Why 4 | Why 5 |
|---|---|---|---|---|---|
| (F) Erroneous information about electricity and water service | Errors in the feasibility of the companies. | Lack of thoroughness in the review of feasibilities. | Lack of expertise of those who evaluated and provided the information. | Failure of internal service procedures. | Limitations of external entities. |
| (G) Irregularities in the background of the land | Error from entities that delivered the land. | Lack of neatness in the elaboration of loans. | Lack of experience from those who developed the loan. | Lack of prior knowledge of the mistakes that could be presented. | Limitations of external entities. |
| (H) Community opposition | Lack of community information about the project. | Lack of dissemination of the project with the community. | Lack of coordination between the municipality, the community, and JUNJI. | Lack of neatness in the public relations of both entities with the community. | Limitations of external entities. |
| (I.1) Design changes due to a change in government regulations | Changes in the regulatory plan. | Lack of knowledge about the regulatory plan change process. | Lack of expertise of the person who requested and delivered the regulatory information. | Lack of technical knowledge. | Lack of experience (JUNJI). |
| (I.2) Design changes on owner's request | Design improvements. | Improvements in project conditions. | Lack of neatness in the review of the project in the preliminary phase. | Lack of communication between user-designers | Lack of experience (JUNJI). |
| (I.3) Design changes due to mechanical and topographical information | Changes in architecture and material design by new calculations. | Lack of information on soil mechanics and topography. | Because they were not allowed to include them at the time of tendering. | For benefiting from exceptional project formulation. | Regulatory limitations of the legal framework for public projects. |
| (I.4) Design changes due to problems with land quality | Ignorance of soil type. | Lack of soil mechanics and topographic information. | Because they were not allowed to include them at the time of tendering. | For benefiting from exceptional project formulation. | Regulatory limitations of the legal framework for public projects. |
| (J) Delay in getting a building permit | Delay in file entry. | Errors in the delivery of information from the file. | Delay in the Municipal Public Works Department review. | Lack of control over the times in the directions of works. | Limitations of external entities. |

**Table 6.** Risk assumption according to the clauses of case study 1.

| Type of Risk | Symptoms | Procurement Documents Section | Contract Clause | Responsible Party |
|---|---|---|---|---|
| Construction risks | (A) Labor force shortage<br>(B) Non-payment claims<br>(C) Materials shortage | 37. Compliance with labor and social security obligations. | Second: contractor's obligation | General Contractor |
| Financial risks | (D) Lack of advance payments | - There is no clause in the procurement documents or contract that relates to this risk, but it is considered that it is the General Contractor that is responsible as he/she determines whether or not to request an advance payment | | Owner |
| | (E) Delay of payments of the client handing over the client | - There is no clause in the procurement documents or contract that relates to this risk, but it is considered that it is the Client who is responsible, given the characteristics that define it | | |

**Table 7.** Risk assumption according to the clauses of case study 2.

| Type of Risk | Symptoms | Procurement/Contract Clause | Responsible Party |
|---|---|---|---|
| Management risks | (F) Erroneous information about electricity and water service.<br>(G) Irregularities in the background of the land.<br>(H) Community opposition. | 35. Term and duration of the contract: paragraph 8 | Owner |
| Construction risks | (I.4) Design modification due to problems with land quality | 35. Term and duration of the contract: paragraph 8 | |
| Political risks | (I.1) Design modification due to a change in government regulations | 35. Term and duration of the contract; paragraph 6 | |
| Design risks | (I.2) Design modification on owner's request | | |
| Design risks | (I.3) Design modification for mechanical and topographical information | 36. Variation in quantity | General Contractor |
| Third-party and complexity risks | (J) Delay in getting a building permit | 2. Contract execution stage "design of specialties" | |

## 5. Discussion

The results show that most of the ECT projects presented labor force shortage, non-payment claims, and materials shortage. The root-cause analysis helped associate the symptoms directly with deficiencies in the bid evaluation process, the lack of JUNJI's experience, and the regulatory limitations of the legal framework for public projects.

The deficiencies in the bid evaluation process to select the general contractor as the root cause of early contract termination are consistent with the literature considering several authors have highlighted the importance of proper general contractors for project success [7,42,43]. According to the interviewees, the selection process was not sufficiently rigorous to corroborate the experience, technical competence, and financial capacity of those who were hired. On the other hand, considering that the general contractor assumes the construction stage risks, the owner cannot employ strategies to improve the project's progress. This is because the JUNJI professionals only control the construction quality and compliance with the legal framework regulations, which means they do not have control over decisions in the stages where there were ECT symptoms. The latter is related to JUNJI's governance and their lack of experience in building projects and the lack of access to a reliable database that shows qualifications and experience to choose a general contractor with the abilities required to manage a DB contract to develop this type of project.

The deficiencies in the bid evaluation process trigger the selection of a general contractor with a lack of experience. The documentary review shows that the general contractor did not have the technical and administrative competence to manage a design–build contract, which requires bearing all risks in the construction stage and most of the risks in the design stage. The lack of experience is reflected by financial difficulties and delays in payment to subcontractors due to deficiencies of the general contractor managing the financial resources. Consequently, the results expose that all projects presented delays and deviations from the original schedule due principally to labor force shortage, non-payment claims, and materials shortage.

The lack of JUNJI's experience and regulatory limitations of the legal framework for public projects also act as a root-cause of early contract termination. JUNJI focuses on the administration of kindergarten education to provide quality education and comprehensive well-being to children, preferably between 0 and 4 years of age. Hence, JUNJI's experience does not focus on managing construction projects compared to other government institutions (e.g., the Ministry of Public Works). Its infrastructure department did not have experience with this type of contract (design–build) nor managed many projects with simultaneous processes (bidding and construction). For that reason, JUNJI created a new independent department to manage the program's requirements and design–build contracts. However, this contract to the program "Aumento de Cobertura" was the first JUNJI carried out because they traditionally used the project delivery design–bid–build approach. In addition, according to the interviewers, JUNJI does not have a contractors' registry or an adequate system to manage the distribution of the project budget, incentives, and payments of the general contractor. Hence, the lack of JUNJI's experience and regulatory limitations of the legal framework for public projects were evidenced by the financial difficulties and delays in payments for completed work during the construction phase, which does not provide sufficient and stable cash flow.

The completed projects were also subject to contract modifications in terms of costs and schedules. The results show that these modifications are due to the management of external service and design modifications. It is presumed that the projects were completed because the symptoms identified were associated with JUNJI's risks. The owner could process budget adjustments within acceptable regulatory ranges to conclude the project and manage risks involving external services and public relations with the corresponding entities.

Likewise, it was identified that among the root causes of the ECT and completed projects, only JUNJI's lack of experience was common. However, despite JUNJI's lack of experience with projects of these characteristics, this cause, by itself, would not be a

determining factor in presenting ECT. Nevertheless, it can impact by itself the project performance in terms of cost and schedule.

Based on the above, two critical factors were identified to cause ECT. The first is a poor choice of the general contractor, especially for the type of contract employed in the program "Aumento de Cobertura." The legal framework of the tender documents and regulatory limitations did not allow the owner to correctly evaluate the general contractor's technical competence and financial capacity. The other critical factor is JUNJI's lack of experience and governance in managing the design–build contract. JUNJI did not have an adequate control system that allowed the early detection of consequences to apply mitigation strategies and prevent ECT.

To mitigate the impact of the critical factors identified, modifying the tendering process and the management and control system is recommended. First, it is necessary to include a more precise evaluation of the general contractor's experience and financial capacity. Specifically, it is suggested to create a registry of general contractors, considering that JUNJI gained experience executing around 350 kindergartens projects throughout the country. It is also recommended to define a better methodology for management and control by JUNJI which would allow them to make timely decisions to improve the work's progress and not just follow recommendations from ITO. Furthermore, JUNJI's responsibility regarding the timely availability of funds must be properly regulated to avoid higher costs due to delays not attributable to the general contractor. It is also suggested to improve the mechanism of incentives and control of the project budget to maintain the general contractor's interest in successfully closing the project.

In summary, the case studies' evaluation reveals the lack of management strategies to deal with the potential consequences of the design–build contract's risks, which clearly were not mitigated correctly in the case study projects. Furthermore, the results expose that it only takes a symptom (A)–(C) (e.g., unit of analysis 7) or two ((A)–(E) (e.g., unit of analysis 2)) to lead to severe breaches of the contract, which allow the owner of the contract to require ECT.

## 6. Conclusions

This study addressed one of the most severe problems in public infrastructure projects in developing countries: early contract termination. This work analyzed the main factors that lead to this condition through a multi-case study. The evaluation of twenty kindergarten construction projects demonstrated that the main causes of ECT are deficiencies in the bidding evaluation process, which led to the selection of the general contractor with a lack of experience, lack of experience of the owner, and regulatory limitations of the legal framework for public projects. The results confirm that the ECT condition was a consequence of poor risk management principally by the general contractor. The pattern matching results show that all the ECT and completed projects presented symptoms associated with different risk types. This is reflected in consequences such as scheduling deviations and cost overruns.

The symptoms identified for the analysis units of case study 1 are different from those of case study 2. On the one hand, the symptoms found in case study 1 are labor force and materials shortage, non-payment claims, lack of advance payments, and delayed payments by the client. The root-cause analysis determined that these risks are construction and financial risks and that most of them are the general contractor's responsibility. On the other hand, in case study 2, the following symptoms were common: errors in electricity and water information, irregularities in the land background, community opposition, design changes, and delay in getting the building permit. These symptoms were related to risks of design, construction, management, and third-party and complexity risks. Additionally, it was determined that most of these are the responsibility of the owner.

Based on the results, the root cause of the most critical factors that lead to ECT is the general contractor's lack of experience. In DB contracts, where most of the risks are assumed by the general contractor, poor management of them will significantly impact the

performance of the projects. Therefore, the general contractor's lack of experience in the project's execution reflects a deficiency in the procurement process for this type of project delivery method. For this reason, it is recommended to improve the tender documents that regulate the process of choosing the general contractor in the bidding stage.

This study contributes to the literature by analyzing the root causes of the critical factors in public construction projects that lead to early termination of contracts in developing countries. Literature about risks and project performance presents the problems or critical factors that produce time and cost overruns. However, these studies did not address the connection of these problems with the ECT condition. Additionally, aside from a few studies [11], there is no significant literature on the ECT condition in developing countries. These studies provided critical factors for project abandonment or ECT, but they did not analyze the root causes. In contrast, this study analyzed the root causes of the critical factors (symptoms) that lead to the ECT condition.

The diagnosis performed in this study allowed the authors to generate a set of recommendations to improve the contractual documents designed for this type of project. However, this work's scope is limited since only JUNJI (Ministry of Education) projects were analyzed, and no samples were taken from other entities. Because of this, the objectives and the research question are answered within the framework of the sample. Therefore, it is recommended for future works to consider an intersectoral sample, considering projects from different ministries and institutions. Besides, it is suggested to consider a greater number of units of analysis and other criteria for their selection. For example, choosing projects in the same region and developed simultaneously, since the context under which each project is developed greatly influences the results.

**Author Contributions:** Conceptualization, C.R., H.A.M., A.L.R. and J.A.G.; Methodology, C.R., H.A.M., J.A.G. and A.L.R.; Validation, C.R., H.A.M., J.A.G. and A.L.R.; Formal analysis, C.R., H.A.M., J.A.G. and A.L.R.; Investigation, C.R. and H.A.M.; Data curation, C.R.; Writing—original draft preparation, C.R. and A.L.R.; writing—review and editing, H.A.M. and J.A.G.; Supervision, H.A.M.; Funding acquisition, H.A.M. All authors have read and agreed to the published version of the manuscript.

**Funding:** This research was funded by Programa de Inserción Académica (2017) of Pontificia Universidad Católica de Chile.

**Data Availability Statement:** Some or all data, models, or codes that support the findings of this study are available from the corresponding author upon reasonable request. Some or all data, models, or code generated or used during the study are proprietary or confidential in nature and may only be provided with restrictions.

**Acknowledgments:** The authors gratefully acknowledge the experienced professionals and government ministries who shared the case studies' information and knowledge.

**Conflicts of Interest:** The authors declare no conflict of interest.

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
