# Peer review of "Critical Factors Influencing Early Contract Termination in Public Design–Build Projects in Developing and Emerging Economies"

_buildings, doi:10.3390/buildings12050614_

Round 1
Reviewer 1 Report
Some suggestions for supplementing or editing the content of the article:
- Figure 1 is now very simple; Figure 1 might be modified in such a way as to add a more precise account of how the methodology was used.
- Authors should indicate the number of interviews conducted with two key informants the legal coordinator and the project coordinator.
- Authors should consider whether the results of the study, as given in Tables 6 and 7 (risk assumption according to the clauses of both case studies), could be presented with a graphical representation that would clearly show the shares of responsibility for risks.
Some suggestions for minor formatting modifications:
- Arrange the text and figures to fit on the pages and to avoid blank
spaces (e.g. pages 6, 10, 13).
- Arrange the text so as for example Table 1 is not divided into two pages.
- Throughout the article it is necessary to unify the font (according to the instructions) for example in Table 5, column Why 5; edit uppercase / lowercase style e.g. in lines 483-485.
Reviewer 2 Report
In your next papers on the same subject be good to present the schematic model of management/organization of public projects in developing countries. Kind of a "task list" especially for the pre-execution phase, with respect to yur conclussions presented here.
